# Periodic mRNA synthesis and degradation co-operate during cell cycle gene expression

Philipp Eser[1], Carina Demel[1], Kerstin C Maier[1], Björn Schwalb[1], Nicole Pirkl[1], Dietmar E Martin[1], Patrick Cramer[1,2,*] & Achim Tresch[3,4,**]

## Abstract

During the cell cycle, the levels of hundreds of mRNAs change in a periodic manner, but how this is achieved by alterations in the rates of mRNA synthesis and degradation has not been studied systematically. Here, we used metabolic RNA labeling and comparative dynamic transcriptome analysis (cDTA) to derive mRNA synthesis and degradation rates every 5 min during three cell cycle periods of the yeast *Saccharomyces cerevisiae*. A novel statistical model identified 479 genes that show periodic changes in mRNA synthesis and generally also periodic changes in their mRNA degradation rates. Peaks of mRNA degradation generally follow peaks of mRNA synthesis, resulting in sharp and high peaks of mRNA levels at defined times during the cell cycle. Whereas the timing of mRNA synthesis is set by upstream DNA motifs and their associated transcription factors (TFs), the synthesis rate of a periodically expressed gene is apparently set by its core promoter.

**Keywords** cell cycle; gene regulation; mRNA degradation; mRNA turnover; periodic transcription

**Subject Categories** Transcription; Quantitative Biology & Dynamical Systems

**Mol Syst Biol. (2014) 10: 717**

## Introduction

The eukaryotic cell cycle is controlled by periodic gene expression. Gene expression changes during the cell cycle have been studied extensively in the budding yeast *Saccharomyces cerevisiae* (Wittenberg & Reed, 2005), and in the fission yeast *Schizosaccharomyces pombe*. These studies have revealed transcriptional regulatory proteins that drive cell cycle progression, their DNA-binding motifs, and their target genes (Simon *et al*, 2001; Lee *et al*, 2002; Pramila *et al*, 2006). Parts of the regulatory networks that drive periodic

gene expression could be reverse engineered (Wu *et al*, 2006; Hu *et al*, 2007). Cyclin-dependent kinases (CDKs) are pacemakers of the cell-cycle oscillator (Haase & Reed, 1999; Lu & Cross, 2010), although the sequential expression of TFs is sufficient to produce periodic expression for many cell cycle genes in the absence of mitotic cyclins (Orlando *et al*, 2008). A model suggesting the coupling of a TF network to CDK activity for robust oscillations in the cell cycle has been proposed (Kovacs *et al*, 2012).

The basis for these discoveries was laid by measurements of gene expression along the cell cycle, followed by identification and quantification of cell cycle regulated genes (Cho *et al*, 1998; Spellman *et al*, 1998; Rustici *et al*, 2004). Different studies have identified diverse sets of 300–1500 genes that are periodically expressed (Spellman *et al*, 1998; de Lichtenberg *et al*, 2005b; Granovskaia *et al*, 2010) [for a comprehensive overview of the results of different studies see the Cell Cycle database (Gauthier *et al*, 2008)]. The variation in the total number and the overlap of reported cell cycle genes arises from variation in experimental conditions like synchronization, strain, technological platform, and the type of computational analyses (de Lichtenberg *et al*, 2005a). There are two principal approaches to the identification of periodically expressed genes, non-parametric (model-free) approaches (Spellman *et al*, 1998; Wichert *et al*, 2004) and parametric (model-based) methods (Tavazoie *et al*, 1999; Johansson *et al*, 2003; Lu *et al*, 2004; Guo *et al*, 2013). A successful screening method needs to account for measurement noise and outliers, and ideally provides a smoothed, error-corrected estimate of the expression time course (de Lichtenberg *et al*, 2005a). Additionally, it has to account for the loss of synchronization of cells along the time course, which is caused by variability in progression through the cell cycle.

The regulation of mRNA levels not only involves changes in mRNA synthesis but also changes in mRNA degradation (Tavazoie *et al*, 1999). Periodically expressed genes are enriched among genes that are subject to cytoplasmatic capping which might also contribute to controlling mRNA stability in the cell cycle (Mukherjee *et al*, 2012). Recently, long non-coding (lnc) RNAs have been found to modulate cell cycle transcription and post-transcriptional events by

1 Gene Center and Department of Biochemistry, Center for Integrated Protein Science CIPSM, Ludwig-Maximilians-Universität München, Munich, Germany
2 Department of Molecular Biology, Max Planck Institute for Biophysical Chemistry, Göttingen, Germany
3 Institute for Genetics, University of Cologne, Cologne, Germany
4 Max Planck Institute for Plant Breeding Research, Cologne, Germany
*Corresponding author. Tel: +49 89 2180 76951; Fax: +49 89 2180 76998; E-mail: cramer@lmb.uni-muenchen.de
**Corresponding author. Tel: +49 221 5062 161; Fax: +49 221 5062 163; E-mail: tresch@mpipz.mpg.de

associating to the mRNA of cyclin-dependent kinases thereby affecting their stability (Kitagawa *et al*, 2013). mRNA degradation is known to determine cellular mRNA equilibrium levels (Munchel *et al*, 2011), and time-variable mRNA degradation can help in establishing a timely and precise adaption of mRNA levels (Romero-Santacreu *et al*, 2009; Miller *et al*, 2011; Rabani *et al*, 2011). Single-cell, single-molecule studies identified the mitotic genes CLB2 and SWI5 for which the process of periodic mRNA synthesis is corroborated by time-delayed periodic fluctuations in the degradation of their transcripts (Gill *et al*, 2004; Trcek *et al*, 2011). Periodically expressed transcripts often encode proteins that are needed at a specific time of the cell cycle (Jensen *et al*, 2006; Yu, 2007). Therefore any mechanism that sharpens the temporal profile of a periodically expressed mRNA is potentially beneficial.

Despite these efforts, major questions concerning cell cycle gene expression remain. First, how do mRNA synthesis rates for periodically expressed genes change during the cell cycle? Second, what are the mechanistic determinants for the timing and magnitude of these synthesis rate changes? Third, do mRNA degradation rates also change during the cell cycle, and if so, how do these changes contribute to the observed changes in mRNA levels, i.e. transcript abundance? Here we tackle these questions by metabolic labeling of newly transcribed mRNA and microarray profiling. This method has been shown to be more sensitive than standard transcriptomics for monitoring changes in the transcriptome, for example after osmotic stress (Miller *et al*, 2011). To estimate absolute mRNA synthesis and degradation rates from such data, we previously developed the cDTA (comparative Dynamic Transcriptome Analysis) protocol (Sun *et al*, 2012).

Here we apply cDTA to synchronized *S. cerevisiae* cells, to monitor mRNA metabolism during three complete cell division cycles in two independent replicates. To analyze the data, we derive a novel model-based screening method which for each gene calculates a periodicity score ranking genes according to their periodic expression. The method overcomes limitations of previous algorithms and extracts biologically meaningful gene-specific and global parameters that characterize the periodic profiles of periodically expressed genes. Our simultaneous measurements of total and newly transcribed (labeled) mRNA provide a high-quality data set that allows for the first time for a systematic analysis of the dynamics of mRNA synthesis and degradation during the cell cycle. With the use of a new dynamic model that estimates changes in mRNA synthesis and degradation rates, we demonstrate that most periodically expressed transcripts show non-random periodic changes in their degradation rates that lead to sharper and higher mRNA expression peaks. Our study provides the first evidence for variable mRNA degradation as a ubiquitous phenomenon that can shape periodic gene expression.

# Results

## cDTA monitors mRNA synthesis and degradation during the cell cycle

To measure mRNA synthesis rates over the yeast cell cycle, we synchronized cells using alpha factor as described (Granovskaia *et al*, 2010) and verified synchronization by FACS analysis (Materials and Methods). For consistency with prior studies, we generated and used a Δ*bar1* strain of yeast (Materials and Methods). After release of cells in G1 phase we used cDTA (Miller *et al*, 2011; Sun *et al*, 2012) to measure the amount of newly synthesized and total RNA at 41 time points separated by 5 min, covering 200 min, corresponding to three cell cycle periods. At each time point newly synthesized RNA was labeled with 4-thiouracil for 5 min (Fig 1A). Using *S. pombe* as an internal standard, we normalized the labeled and total mRNA fractions across the time series to get absolute expression estimates (Supplementary Information, section 1.2). The entire time series experiment was performed in two biological replicates. Because labeled mRNA levels correlate well with mRNA synthesis rates (Miller *et al*, 2011; Sun *et al*, 2012), these data represent the first genome-wide estimation of mRNA synthesis rates in synchronized cells at different time points in the cell cycle.

We did extensive checks to verify the quality of our data set. First, we calculated pair-wise Pearson correlations between labeled and total mRNA samples (Supplementary Information, Fig 5). Correlations (within labeled respectively total samples) were consistently above 0.93. Strikingly, periodic expression already shows in the samples correlation structure. Samples taken at similar time points in the cell cycle have a higher correlation than samples taken at more distant time points in the cell cycle. This leads to a characteristic tri-band diagonal correlation structure, corresponding to the three cell cycles that we monitored. A principal component plot automatically places consecutive samples in a "cell cycle clock", a clockwise spiral, demonstrating that most variation in the data (>74%) is due to periodic expression fluctuations (Supplementary Information, section 2.1, Supplementary Information, Fig 6). Second, by ignoring the time at which measurements were taken, we use all labeled and total measurements of a mRNA as replicates to calculate a high precision estimate of its (cell cycle averaged) synthesis and degradation rate. We compared these estimates with the most recent estimates in (Sun *et al*, 2012) obtained by the same cDTA technique. Encouragingly, they are in excellent agreement not only on a relative, but also on absolute scale (Supplementary Information, Fig 7). The high number of replicates allowed us to additionally derive an empirical variance estimate. These estimates constitute the so far richest information source on steady state mRNA synthesis and degradation (Supplementary Tables 4 and 5).

## Model-based periodicity screening

The problem of identifying cell-cycle regulated genes from expression data has been addressed by several studies. The proposed methods can be grouped into non-parametric methods (Spellman *et al*, 1998; Wichert *et al*, 2004) and parametric (model-based) methods (Johansson *et al*, 2003; Lu *et al*, 2004). Non-parametric methods do not assume a specific shape of a periodic time course, nor do they make particular assumptions on the distribution of the measurement errors. As such, they are inherently robust. However, they merely provide a measure for ranking genes according to their "periodicity" without extracting information on the actual shape of the gene's time course. Parametric methods explicitly infer the "true" expression time course of a gene as a basis for a periodicity test. A proper modeling of the time course will not only increase the sensitivity of periodicity detection, it will provide valuable additional information for the grouping of periodically expressed

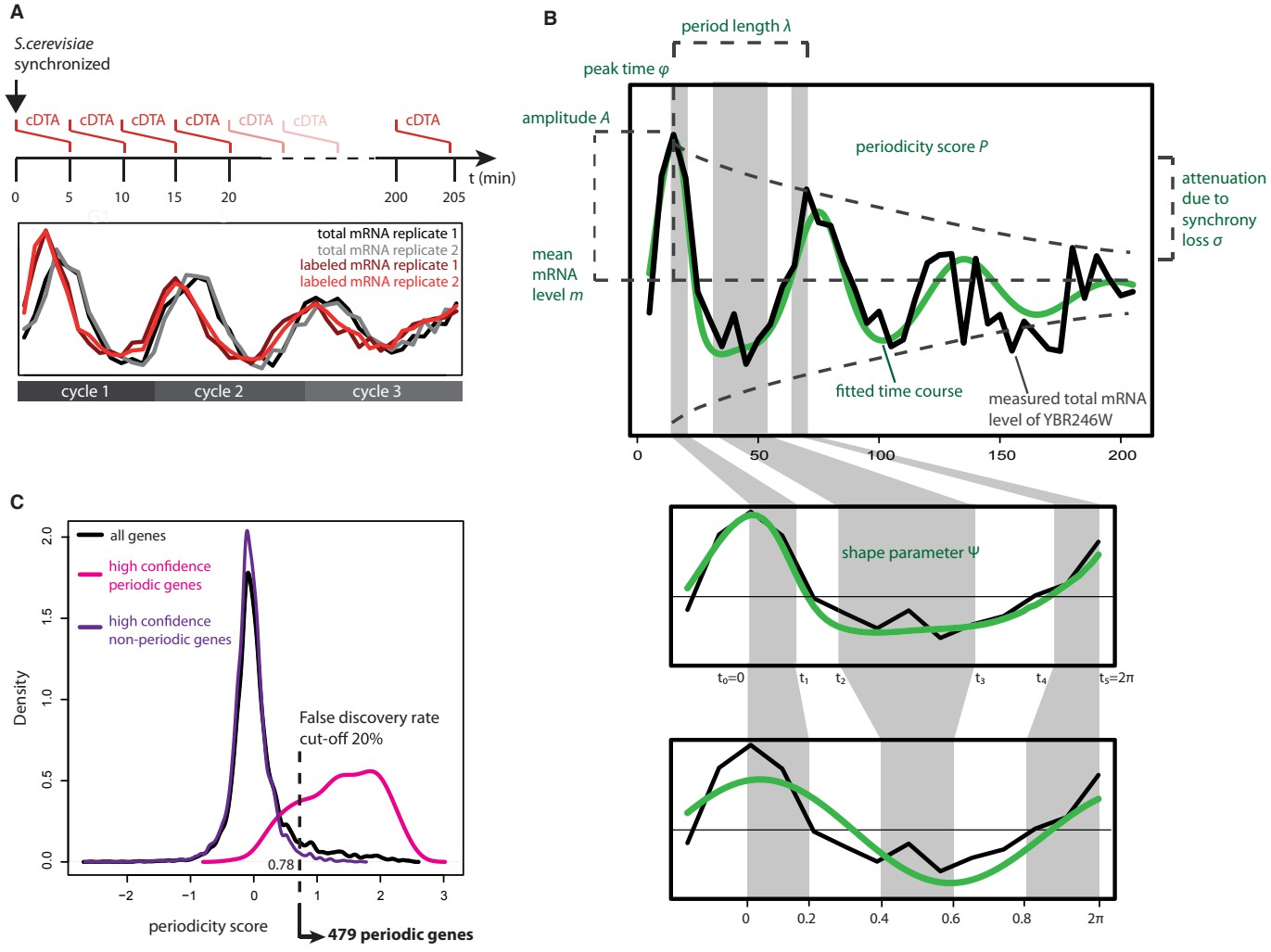

**Figure 1.   Experimental and bioinformatic strategy for the identification and monitoring of periodic mRNA metabolism during the cell cycle.**

A    The cDTA cell cycle time course experiment. mRNAs were labeled with 4-thiouracil every 5 min until $t$ = 200 min. After 5 min labeling time the respective sample was stopped and further processed according to the cDTA protocol. The experiment was performed in two replicates. For each mRNA (YDR400W shown), we obtained two time series of total mRNA levels (black and grey lines), and two time series of labeled mRNA levels (dark red and light red lines).

B    Parametric modeling of periodic time courses by MoPS. Top panel: the peak time ρ denotes the time of maximum mRNA level relative to the time of transcription release. The period length λ is the shortest interval after which the expression pattern repeats, $m$ and $A$ denote the mean mRNA level and the amplitude, respectively. The decrease of the amplitude along several cell cycles is due to synchrony loss, σ. Bottom panel: The cosine wave, our basic model of periodic expression, is adapted to the time series by the shape parameter ψ, which is a monotonic transformation of the "clock" that ticks along the interval [0, 2π].

C    Statistical test for the identification of periodic transcripts. The distribution of periodicity scores (black distribution) is approximated as a mixture of the periodicity score distributions of a set of bona fide periodic (pink distribution) and non-periodic genes (purple distribution). Based on this fit, the 20% false discovery rate cutoff is calculated as 0.78.

genes. On the other hand, parametric models involve the risk of over-fitting, leading to a low specificity in the periodicity test. A careful choice of an appropriate model for periodic gene expression with a sparse parameter set is therefore essential.

Bearing this in mind, we developed a new parametric screening method for periodically expressed transcripts that we call Model-based Periodicity Screen (MoPS; Fig 1B, Materials and Methods, and Supplementary Information, section 1). MoPS is available as an R/Bioconductor package at www.bioconductor.org (can at present be downloaded from www.treschgroup.de/mops.html). To each labeled and total mRNA expression time course, MoPS calculates a

likelihood ratio statistic that compares the best fit of a periodic expression curve to that of a non-periodic curve (Supplementary Information, section 1.1). Periodic expression is modeled by a dampened, deformed cosine wave using six parameters (Fig 1B, Supplementary Informtation, section 1.3). The cell cycle length λ (min) corresponds to the time difference between the first expression peak at peak time φ when the maximum mRNA level is observed and the next expression peak. The periodically changing mRNA level is described by its mean $m$ and its amplitude $A$. The decrease of the mRNA level amplitude with time due to progressive loss of synchronization between cells is described as the "synchrony

loss" σ. We explain this effect by variation in cell cycle length of individual cells in our synchronized population. The parameter σ describes the dispersion of the cell cycle length distribution. The deformation of the cosine wave is described by a "shape" parameter ψ, a bijective transformation of the interval [0, 2π] (Fig 1B, Materials and Methods). Since the mean cell cycle length and the loss of synchronization are a characteristic of the cell population, these two parameters are common to all examined transcripts. This reduces the number of fitted parameters for individual gene expression profiles to four, which makes MoPS extremely robust, despite its flexibility that ensures excellent fits (Supplementary Information, Figs 13 and 16). Like other periodicity tests (Lu *et al*, 2004; Guo *et al*, 2013), our approach can be formulated conveniently as a kernel regression problem (Supplementary Information, section 1.4). For each time point, the kernel function returns the distribution of the cell cycle phases in the population (Supplementary Information, Fig 3). As such, it measures the synchronization respectively the loss of synchronization.

We performed a maximum likelihood fit of each gene expression profile, and then calculated a "periodicity score" $P$ that is defined as the log-likelihood ratio of the best periodic MoPS fit over the best non-periodic fit to an exhaustive set of non-periodic test functions (Supplementary Information, section 1.5 and Fig 1). The magnitude of expression changes is another criterion for improving the detection of periodic transcripts (de Lichtenberg *et al*, 2005b). Our periodicity score uses an expression-dependent (heteroscedastic) error model that implicitly penalizes genes with a low overall expression, since measurements of low abundance genes have a larger (relative) error (Rocke & Durbin, 2001; Supplementary Information, section 1.2).

### Characterization of periodically expressed genes

To identify genes that are periodically expressed, we applied MoPS separately to total and labeled mRNA from both replicate cDTA time series (Supplementary Information, section 2). The cell cycle length λ and the synchronization loss σ were estimated for each gene. The distribution of obtained cell cycle lengths λ sharply peaks at a median of 62.5 min, and the distribution of the synchrony losses σ has a median of 7 min (Fig 2A). The agreement between replicates and between total and labeled mRNA was excellent (Supplementary Information, Fig 11), and our cell cycle length estimate agrees well with that of 65 min in (Granovskaia *et al*, 2010) who used the same strain and the same synchronization method. In a second step, we fixed the parameters λ = 62.5 min and σ = 7 min, and recalculated all other parameters, namely the phase of expression, the character-

istic shape of its time course and the periodicity score for all genes. The obtained values were in excellent agreement between replicates, and also in good agreement between labeled and total mRNA (Supplementary Information, Figs 12–15).

Genes were then ranked according to their periodicity score (for a representative selection of genes and their periodicity scores see Supplementary Information, Fig 9). A cut-off value was chosen based on gold standard sets of periodically and non-periodically expressed genes, to control the false discovery rate at a 20% level (Fig 1C, Materials and Methods). The power of our screening method was increased by combining the periodicity scores obtained from the total and labeled mRNA from both replicates into one sum. MoPS identified a total of 479 periodic genes with high confidence. In the literature, different periodicity screening methods yield between 300 and 1500 genes that are considered cell cycle regulated in yeast (Spellman *et al*, 1998; de Lichtenberg *et al*, 2005b; Granovskaia *et al*, 2010). The agreement between the datasets/methods is moderate (de Lichtenberg *et al*, 2005a), (Supplementary Information, Fig 24), but nevertheless highly significant ($P < 10^{-10}$ in all pair wise Fisher tests). This shows that the detection of periodic genes strongly depends on the method, the experimental conditions, and the stringency cut-off that has been applied. We included MoPS into the benchmark studies in (de Lichtenberg *et al*, 2005a) to verify its performance. A receiver operating characteristic (ROC) analysis showed that MoPS performed as well as state-of-the art methods for the identification of periodically expressed genes (Supplementary Information, section 2.9 and Fig 25). Recall that our primary goal was not the development of a screening procedure which outperforms all other methods in terms of sensitivity and specificity. We rather wanted to have a generative model for periodically expressed genes whose parameters have an intuitive meaning. MoPS' parameters are immediately accessible to biological interpretation, which is of immense practical value as will be demonstrated below.

### Three expression waves during the cell cycle

We sorted all periodically expressed genes by their synthesis peak time (Fig 2C). Among the periodically expressed genes were many prominent cell cycle genes (Murakami & Nurse, 2000) including all six genes of the minichromosome maintenance family (MCM2-7), cyclins, and histone genes. These genes were used to assign cell cycle phases G1, S, G2, and M to measurement time points in our data (Supplementary Information, section 2.7 and Fig 22). Periodically expressed genes appeared to be grouped in three expression waves, in agreement with previous observations (Rowicka *et al*,

---

**Figure 2.  Extraction of biological parameters for periodically expressed transcripts.**

A   Identification of the global parameters cell cycle length (λ, *x*-axis) and synchrony loss (σ, *y*-axis). Each gene yields an estimate (λ, σ). The 3D surface plot shows their joint distribution with the medians λ = 62.5 min and σ = 7 min.

B   Fitted time courses of the 479 periodic genes. Each row corresponds to one gene. Time is measured in terms of cell cycle phases G1, S, G2, M (*x*-axis). Genes were sorted according to their peak time, starting with genes peaking in G1 phase. High (low) expression is encoded in red (blue), where expression is taken relative to the gene's mean expression. The histogram on top shows the distribution of the peak times along the cell cycle. The snake plot to the left shows the improvement of the MoPS fit over a fit with a sine wave. Each box in the plot summarizes 15 consecutive genes. The bottom plot shows how the synchronization of cells decreases with time. Each measurement time point is represented by one column, which is a greyscale-coded representation of the individual cell cycle time distribution across the cell cycle. The golden bar marks the central 50% interval of the respective distribution. The dark red dot within the golden bar marks the modes of these distributions.

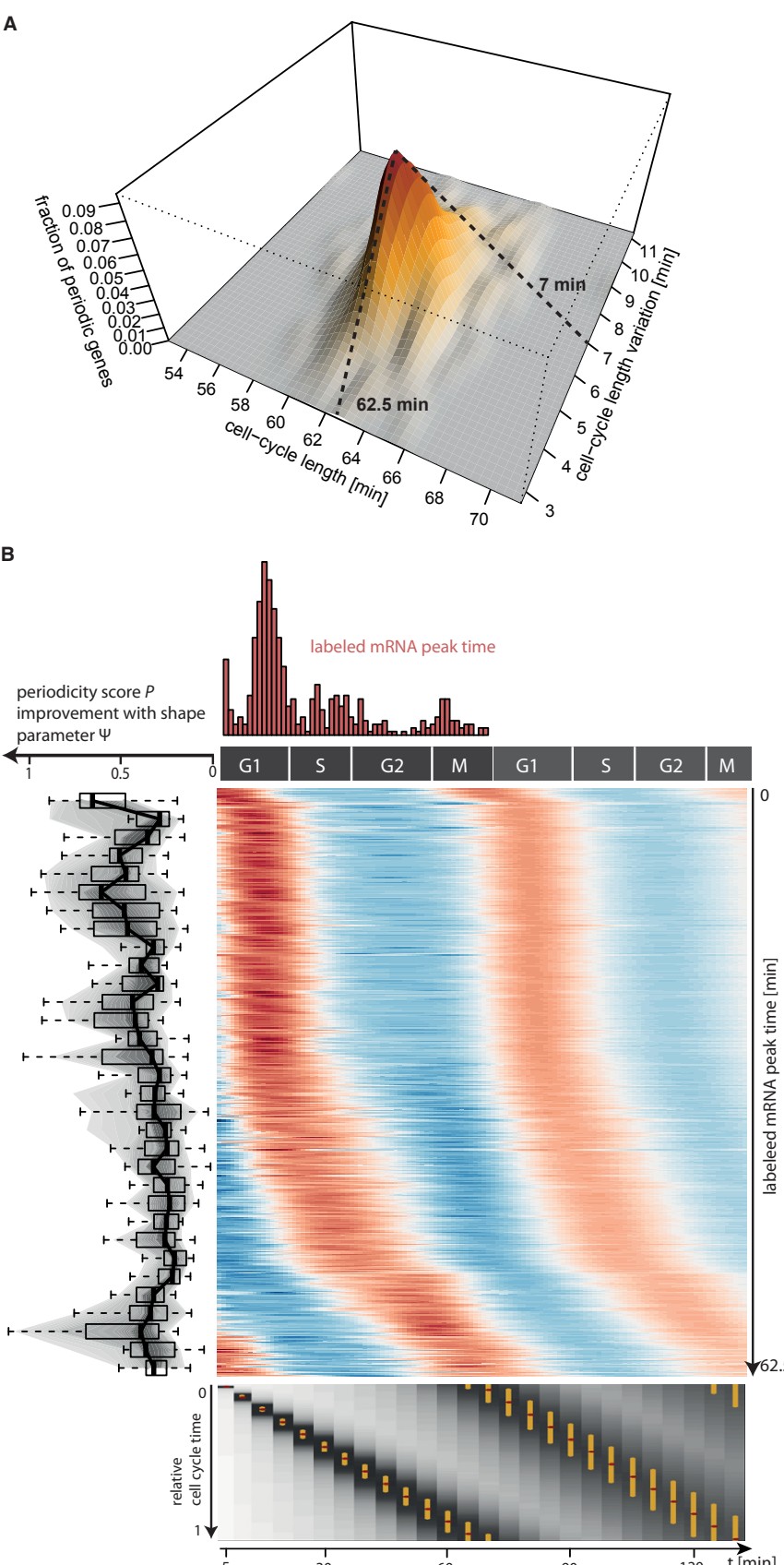

2007). A first wave shows peak synthesis in G1 phase, a second during S phase, and a third at the onset of M phase (Fig 2B).

### Recovery of cell cycle transcription factors

Our 479 periodically expressed genes also contained eight transcription factors (TFs) that potentially regulate the cell cycle. Since there is no consensus set of TFs that regulate the cell cycle we systematically screened for transcriptional regulators of periodic genes (Fig 3A). We used only the labeled mRNA data in this screen, because these represent transcriptional regulation better than total mRNA profiles. The extracted shapes of the periodic genes were grouped into 10 clusters by Euclidean distance average linkage *k*-means clustering. For each of the clusters, we performed an XXmotif search (Hartmann *et al*, 2013) for DNA sequence motifs in a region 500 bp upstream of the experimentally defined transcription start site. In total, 50 motifs with *E*-value smaller than one were recovered. Each motif was then matched to known DNA-binding protein motifs with TOMTOM (Materials and Methods).

We obtained a total of 50 DNA motifs that were associated with a total of 32 DNA-binding transcription factors (Supplementary Table 1). The top motif identified from a G1 cluster perfectly

matched the known Mlu1 cell cycle box (MCB) motif (Fig 3B). The MCB motif is enriched in promoters of genes required for DNA synthesis. TOMTOM identified two TFs that were significantly associated with the MCB motif, MBP1 and SWI6, which form the MBF heterodimer in which MBP1 acts as a sequence-specific, DNA-binding trans-activator. MBF regulates expression during the G1/S transition (Koch *et al*, 1993). The second best motif that was found to be enriched in M phase was matched by multiple TFs (MCM1, NDD1, YOX1, FKH2, DIG1, ASH1, and FKH1), reflecting a complex interaction network of activators and repressors. The repressor Yox1 and the activator Fkh2-Ndd1 compete for binding to Mcm1, although they associate at opposite sides of the dimeric Mcm1 transcription factor. This competition determines the expression of late mitotic genes in yeast (Darieva *et al*, 2010).

The obtained set of 32 predicted cell cycle TFs partially overlaps with TFs in other studies that integrate expression data with motif-discovery tools (Banerjee & Zhang, 2003; Tsai *et al*, 2005; Cheng & Li, 2008; Orlando *et al*, 2008; Wu & Li, 2008; Fig 3C). It is evident that the association of TFs with cell cycle regulation is only clear for a core set of a few TFs (Fig 4C). Wu and Li (Wu & Li, 2008) performed a benchmark on TFs annotated as known cell cycle regulators (Mewes *et al*, 2011) with the Jaccard index as a measure

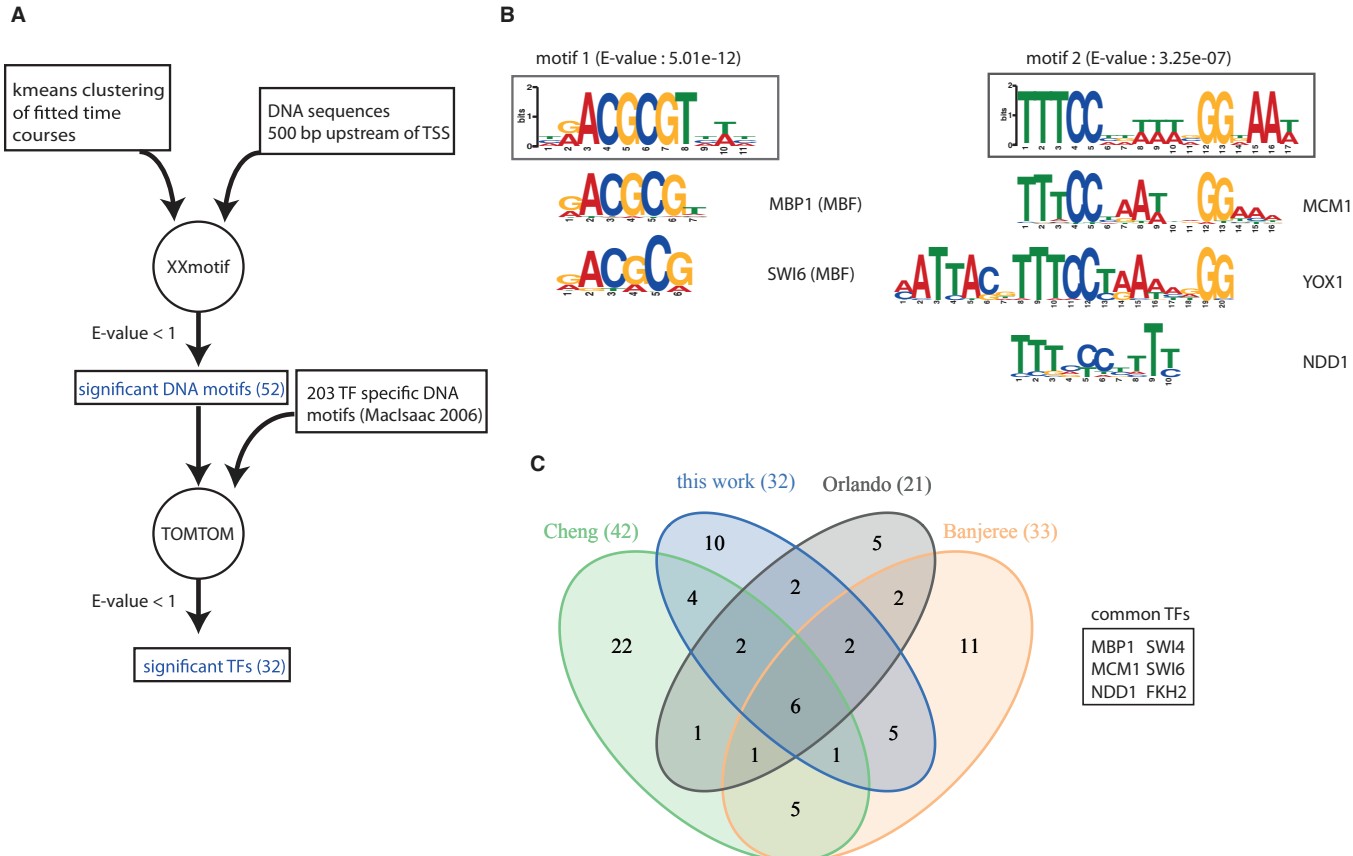

**Figure 3.    Identification of cell-cycle related DNA motifs and transcription factors.**

A    Workflow, consisting of a motif discovery step using XXmotif and a TF detection step using TOMTOM. The inputs to XXmotif are the 500 bp upstream sequences of sets of co-regulated genes. The resulting list of significantly enriched motifs is processed by TOMTOM to find TFs with matching binding sites.

B    Sequence logos of the two top motifs and their associated TFs (MBF for motif 1; MCM1, YOX1 and NDD1 for motif 2) together with their *E*-value.

C    Venn diagram showing the overlap of various integrative bioinformatic methods for the prediction of cell-cycle related TFs.

                    

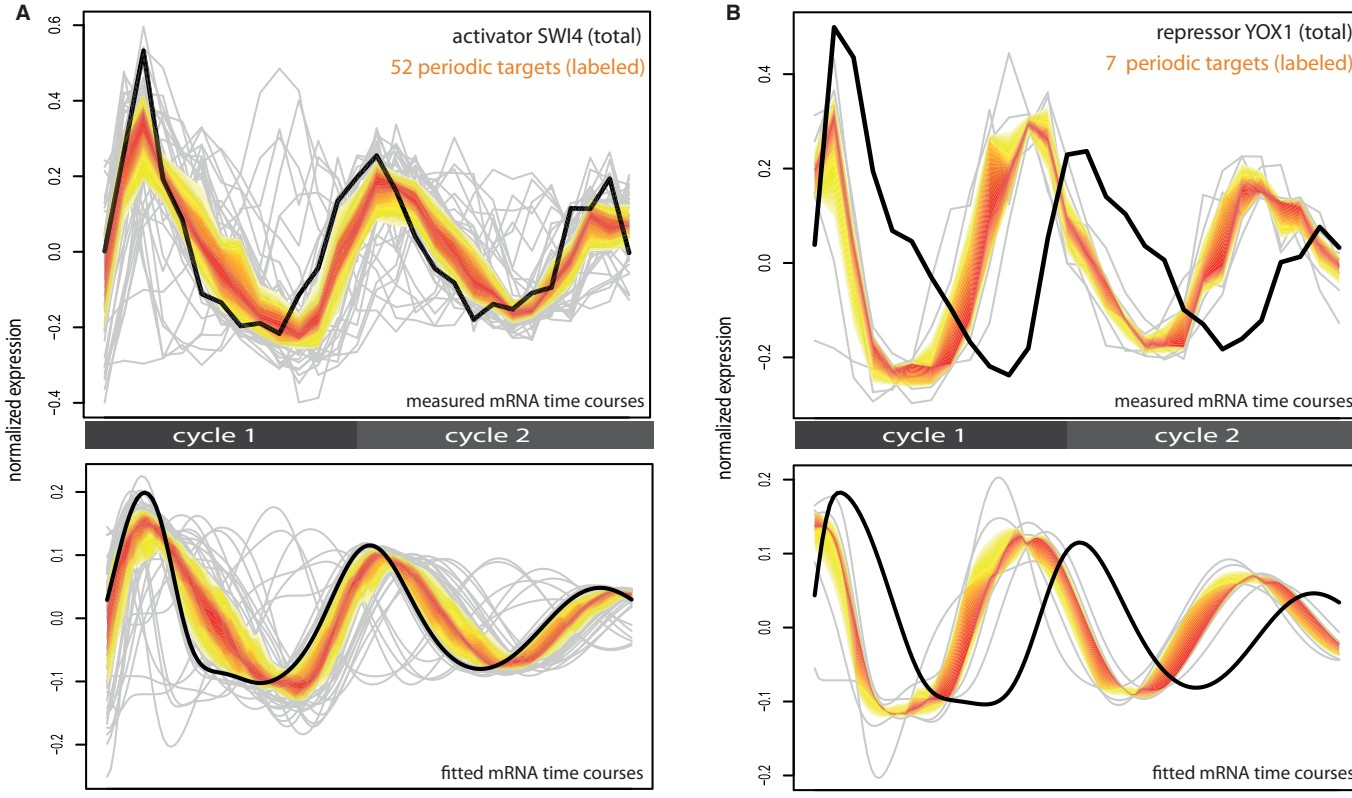

**Figure 4.   Cell cycle regulators and their target genes.**

A, B    Two examples of identified cell-cycle regulating TFs [SWI4 (A), YOX1 (B)] and their periodically expressed target genes. The top panel shows the measured, mean-centered and re-scaled time courses of the TF (total mRNA, black line) and its periodically expressed targets (labeled mRNA, grey lines). The yellow-orange band marks the expression range of the central 50% of the targets at each time point. The bottom panel shows the corresponding fitted time courses as derived from our screening procedure.

of agreement. Their method scores best (Jaccard index 0.293), whereas our set of TFs led to a Jaccard index of 0.275, which is higher than 0.245, the score of the second best TF set (Tsai *et al*, 2005) in their study.

## TFs govern the expression timing of periodic genes

We investigated the influence of cell cycle regulating TFs on the mRNA synthesis of their target genes. Using ChIP-chip derived TF-target gene associations (MacIsaac *et al*, 2006) of our 32 cell cycle regulators to our 479 periodic genes, we compare the total mRNA time course of a TF to the labeled time course of its targets. Eight of our 32 cell-cycle TFs are periodically expressed themselves. Their time course of total mRNA levels corresponds to their regulatory role in cell cycle-associated transcription activation or repression.

The expression of an activating TF is expected to precede the synthesis of its target genes. This is in accordance with our observations. SWI4 is a known activator working together with SWI6 to activate G1-specific transcription of targets. Indeed, the level of SWI4 mRNA peaks shortly before the synthesis peaks of its periodic target genes (Fig 4A). In contrast, the expression of a repressive TF should be preceded by the synthesis peak of its target genes. Indeed, the transcriptional repressor YOX1 that regulates genes expressed in

M/G1 phase (Ubersax *et al*, 2003) shows high expression after peak synthesis of its target genes, and low mRNA levels when the synthesis rate of its targets is high (Fig 4B). The periodically expressed gene FKH2 is described as having a dual role as activating and repressing TF (Sherriff *et al*, 2007). Its targets peak either at the onset of M phase, shortly after the FKH2 peak, or at late G1 phase, shortly before the FKH2 peak. The first group is consistent with an activating role of FKH2, the second group seems to be repressed by FKH2 (Supplementary Information, Fig 28). Targets of non-periodically expressed TFs show also coherent timing, the most compelling example being the TF MBP1 and genes exclusively targeted by MPB1 (Fig 5A). The same effect was observed for all target gene sets with identical motif composition in their upstream region (Supplementary Information, Fig 29). Thus, in many instances the expression levels of regulatory TFs could explain the synthesis rates of their target genes.

## The core promoter governs the synthesis rates of periodic genes

The mean and amplitude of periodic genes are highly correlated (Pearson correlation $r > 0.81$, Supplementary Information, Fig 20). The distribution of the mean expression $m$ of the periodic genes is comparable to that of all genes, with the exception of the left tail of

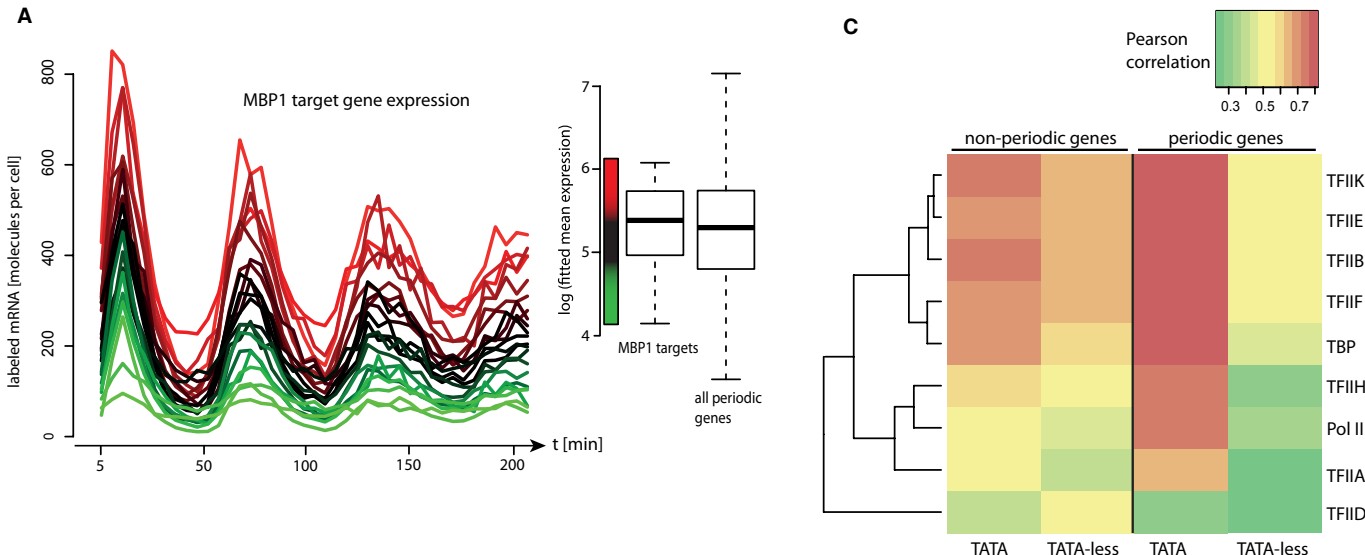

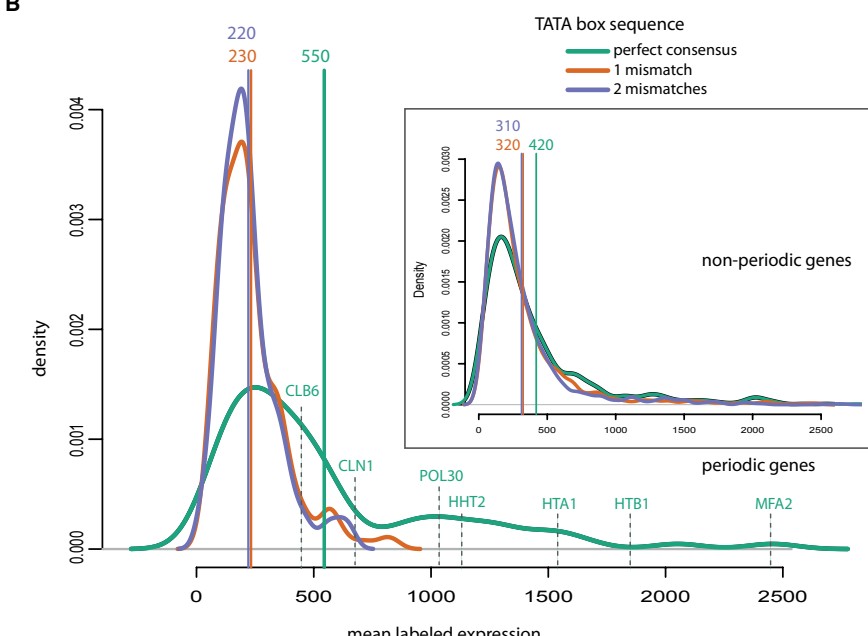

**Figure 5.  Promoter and enhancer structure determine expression strength and timing.**

A    Time courses (absolute labeled mRNA measurements) of 22 periodically expressed genes that are exclusively annotated as MBP1 targets. Colors correspond to mean expression levels extracted from our fitting procedure. Box plots on the right show the expression ranges of the 22 MBP1 targets (left) and all 479 periodic genes (right) in logarithmic scale.

B    Densities of mean labeled mRNA expression of periodic or non-periodic genes, respectively (inset), stratified for the number of mismatches to the TATA consensus motif at core promoter (green: 0 mismatches, orange: 1 mismatch, purple: 2 mismatches). The green, orange and purple vertical lines indicate the mean (rounded to nearest tenth) of the respective distributions. Selected periodic genes that have a consensus TATA-box and are associated with enriched cell-cycle processes (seven out of 80, see text) are marked.

C    Correlations between the mean labeled expression of non-periodic and periodic genes grouped into TATA-containing (0 mismatches) and TATA-less with core promoter occupancies of general TFs involved in pre-initiation complex formation. The heatmap colors range from green (moderate correlation) to yellow (good correlation) and red (high correlation).

weakly expressed genes (Supplementary Information, Fig 21). This is not surprising, because periodic genes fluctuate in their expression, which necessarily leads to a certain minimum mean expression level. The genes exclusively regulated by MBP1, though agreeing well in their timing, showed a remarkable diversity in their synthe-sis mean and amplitude (Fig 5A). The distribution of their mean synthesis rates resembles that of all periodic genes. This could also be observed with other sets of target genes which are regu-lated by common cell cycle TF(s) (Supplementary Information, Fig 29).

This suggested that TFs determine the timing but not the magnitude of the transcription rate of their target genes. We therefore checked whether the synthesis rate is rather set by the target gene core promoter sequence. We analyzed the deviation of the TATA box sequence from the TATA consensus. Genes were partitioned into three groups, genes with a perfect TATA box (0 mismatches to the TATA consensus motif), and TATA-less genes showing one or two mismatches compared to the TATA box consensus at the experimentally defined location where the transcription pre-initiation complex is formed (Rhee & Pugh, 2012). We excluded genes with more than two mismatches from the analysis, since only three of these genes were periodically expressed.

For non-periodic genes the distribution of mean synthesis rates peaked at similar values for all TATA groups, with perfect TATA-containing genes peaking only slightly higher than TATA-less genes ($P$-value $< 1.7^{-5}$, Wilcoxon test; Fig 5B). For periodic genes, however, the perfect TATA box group showed a substantially higher mean synthesis rate than the imperfect TATA box groups ($P$-value $< 10^{-10}$, Wilcoxon test). Although the differences are significant for non-periodic and periodic genes, the effect is threefold stronger for periodic genes. Indeed, periodic genes with very high levels in total and labeled mRNA were almost exclusively found in the perfect TATA box group. Gene Ontology analysis (Bauer *et al*, 2011) of the 80 periodic genes with a consensus TATA box (using all 479 periodic genes as background) showed enrichment for processes of cell cycle progression, with the most significant process being the regulation of CDK activity by cyclins (CLN1, CLN2, CLB1, CLB6, PCL7, and PCL2). Further enriched Gene Ontology categories include DNA replication (POL12, POL30), chromosome organization during meiosis (MCD1, SGO1, GNA1), and chromatin assembly and histone formation (HTB1, HTA1, HHT2; Fig 5B).

To corroborate these findings, we compared the occupancy levels of the general transcription initiation factor TFIIB at core promoters (Rhee & Pugh, 2012) to the mean labeled mRNA levels of periodic and non-periodic genes. We observed a high correlation of TFIIB occupancy with the expression mean (Fig 5C) and amplitude. The highest correlation was found for periodic TATA-box containing genes. Whereas other initiation factors behave like TFIIB, the initiation factor TFIID occupancy correlated only weakly with expression levels of periodic genes, regardless of the core promoter sequence. This is in line with the proposed role of TFIID in the transcription of constitutively expressed genes (Huisinga & Pugh, 2004). To conclude, the mRNA synthesis of cell-cycle regulated genes is governed by the sequence of the core promoter rather than the binding of upstream TFs, which however control the timing of expression.

**Degradation rates of periodic mRNAs are not constant**

Assuming that all copies of a transcript in an mRNA population share the same hazard of being degraded, the time course of an mRNA population is described by the differential equation

$$dT/dt = \mu(t) - \delta(t) \times T \quad (*)$$

where $T$ is the mRNA level, $\mu(t)$ is the time-dependent synthesis rate and $\delta(t)$ is the time-dependent degradation rate for that population. Given $\mu(t)$ and $\delta(t)$, Equation (*) can predict the time course of total and labeled mRNA levels. Note that Equation (*) leaves one degree of freedom, the boundary condition on $T$. By setting $T(0)$ to the total RNA level at time 0, the resulting solution $T(t)$ to Equation (*) is the time course of the total RNA. By letting $T(t_j) = 0$, the solution $T(t)$, for $t > t_j$, is the amount of labeled RNA obtained after a $(t - t_j)$ min labeling pulse starting at time $t_j$. For a description of the numerical and analytical solutions to Equation (*) see (Supplementary Information, section 3.1).

We used Equation (*) to simulate how a peak in mRNA synthesis translates into total mRNA in different degradation rate scenarios (Fig 6A). In particular, we computed the peak time delay between synthesis rate peak and total mRNA peak. A constant, low degradation rate leads to a broad peak in total RNA with a large peak time delay. A constant, high degradation rate reduces this time delay substantially, yet at the expense of a reduced total mRNA level. A variable degradation rate with a peak following the synthesis rate peak however results in a shorter peak time delay while maintaining a high total mRNA peak. The simulation shows that appropriate changes in the mRNA degradation rate minimize the peak time delay while still achieving a quantitatively high total mRNA response.

We therefore compared the peak time in labeled and total mRNA for each periodic gene (Fig 6B). This revealed a median time delay of 2 min between the total RNA with respect to the labeled RNA peak (mean 2.8, 1st quantile 0, 3rd quantile 4.0 min). According to our dynamic model, the expected time delay on the basis of a median transcript half-life of 11.5 min however is 8 min. Assuming constant degradation rates, the observed short peak time delays could only be explained by very short half-lives in the range of 1–2 min. This is far below any estimate in the literature (Miller *et al*, 2011; Munchel *et al*, 2011; Wang *et al*, 2002). For example, the ten cyclins which are found as periodically expressed in our data have a mean peak shift of 0.8 min. The observed short delays between synthesis and total mRNA peaks in periodic transcripts are therefore incompatible with the assumption of constant degradation rates.

**Periodic changes in mRNA degradation shape expression peaks**

To investigate the potential role of mRNA degradation rate changes quantitatively, we extended the DTA method such that it allows for the estimation of changes in mRNA synthesis and degradation rates. We exploit the fact that equation (*) translates a synthesis time course $\mu(t)$ and a degradation time course $\delta(t)$ into predictions of total and labeled mRNA (Fig 6A). This can be used to reverse engineer $\mu(t)$ and $\delta(t)$ from a pair of observed labeled and total mRNA time courses (Fig 6C). We model the synthesis rate as a piece-wise linear function, whereas the degradation rate $\delta(t)$ is modeled as sine function (see Supplementary Information, section 3.2). Note that we did not use the smoothed synthesis rate estimate of MoPS, because MoPS aimed at the detection of periodic expression, and did not take into account changes in mRNA degradation. Moreover, we wanted to exclude any model bias and avoid findings due to slightly biased model assumptions. The measurement error that determines the quality of fit was as in MoPS. The parameters were then fitted to the measured cDTA data by Markov Chain Monte Carlo (Supplementary Information, section 3.2). This enabled us for the first time to decompose cell cycle dependent mRNA expression into the processes of mRNA synthesis and degradation. The rate estimates for all expressed genes are listed in Supplementary Tables 2 and 3.

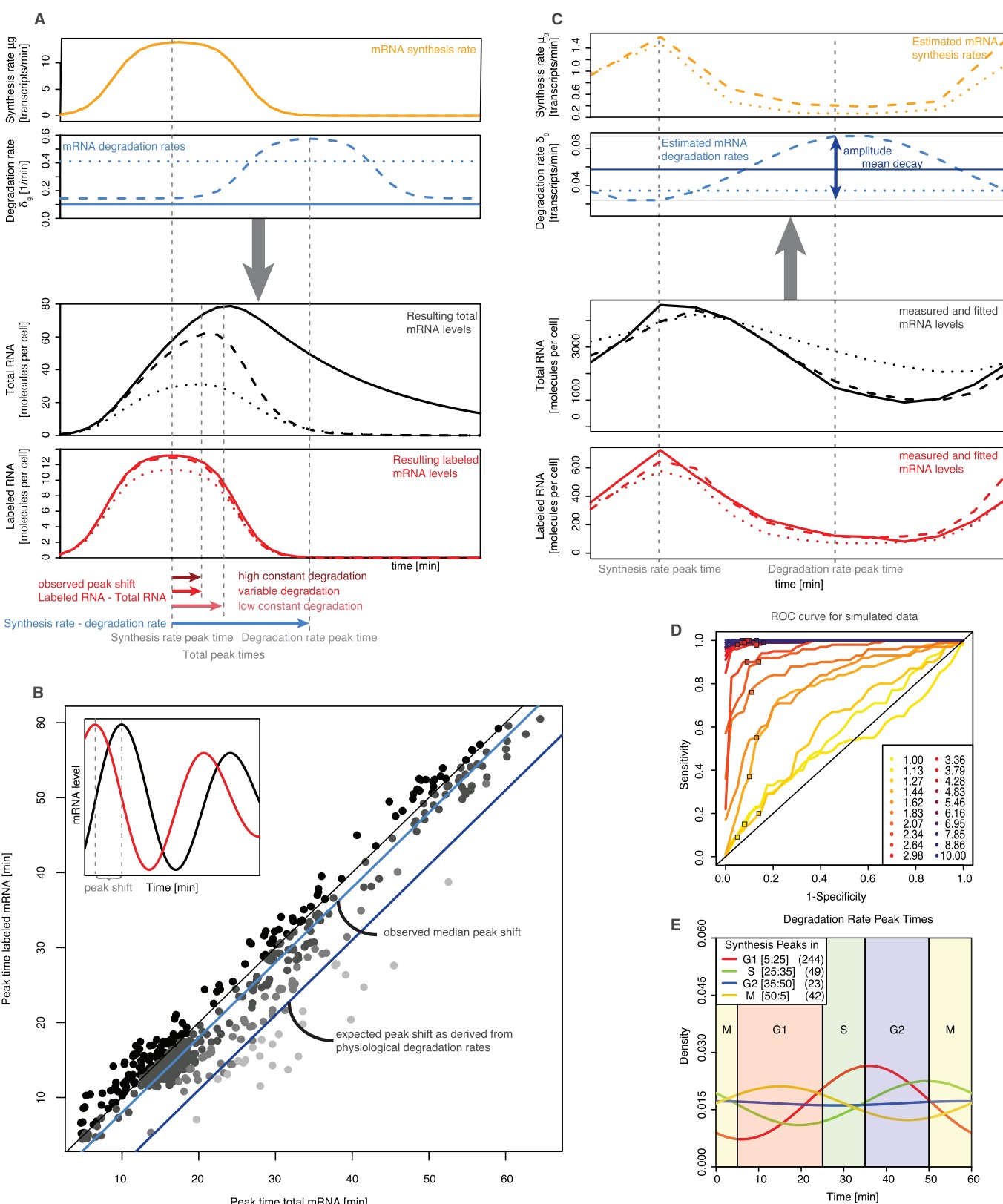

We further developed a score quantifying the strength of periodic mRNA degradation. It is based on the comparison of two models for the explanation of the labeled and total mRNA time series of a gene. One model assumes a constant mRNA degradation rate, $\delta(t) = const$, and the other assumes a sinusoidal degradation rate, $\delta(t) = a*\cos(t - \phi) + const$. The log likelihood ratio of the

**Figure 6.   mRNA turnover model and detection of periodic mRNA degradation.**

A   Calculation of labeled and total mRNA time courses as functions of mRNA synthesis and decay. A peak in RNA synthesis rate (top, orange line) translates into different time courses of total RNA concentration (3rd panel, black lines) and labeled RNA concentration (bottom, red lines) according to different time courses of its degradation rate (2nd panel, blue lines). Shown are three realistic degradation scenarios: The solid or dotted blue line corresponds to a constant low or high degradation rate, respectively. The dashed blue line shows a scenario in which degradation peaks a while after the synthesis rate peak. A low (high) constant degradation rate leads to a long (short) peak shift between total mRNA and RNA synthesis rate. A variable, peaked degradation leads to a short peak shift, while at the same time achieving almost the same amplitude of total RNA variation. The high constant degradation rate was chosen such that the resulting total RNA curve has the same peak shift as in the peaked degradation scenario. This leads to substantially decreased peak amplitude.

B   Scatter plot of labeled versus total RNA peak time for 479 periodically expressed genes. The distance of a point to the main diagonal (thin black line) measures the peak shift for the corresponding gene (see inset, illustrating the peak shift between a total (black) and labeled (red) mRNA time course). Solid light blue line: observed median peak time delay = 2 min, corresponding to a constant mRNA degradation rate of at least 0.4 (half-life of 1.7 min). Solid dark blue line: expected median peak shift = 8 min corresponding to the average degradation rate δ = 0.06 in *S. cerevisiae* (half-life of 11.5 min).

C   Fitting of the dynamic mRNA turnover model to experimental data. The two lower panels show the measured labeled RNA (solid red line) and total mRNA (solid black line) time courses for YNL312W (RFA2). The time courses were fitted using either a variable degradation scenario (red/black dashed lines) or a constant degradation scenario (red/black dotted lines). In both scenarios, the synthesis rates are estimated by a piecewise linear function (first panel, dashed/dotted orange lines) and the degradation rates are estimated by a constant or a sine function (second panel, dashed/dotted blue lines), respectively.

D   Statistical power of the variable degradation score. The receiver operating characteristic (ROC) curves show the variation of sensitivity and specificity as a function of the degradation score. Each curve represents a different simulation scenario, in which the genes with variable degradation had a relative amplitude ranging from 1 (corresponds to constant degradation) to 10.00 (high fluctuation). The solid squares denote, for each scenario, the sensitivity and specificity achieved for a degradation score cutoff of 0.3.

E   Distribution of degradation rate peak times in the cell cycle. The genes are grouped according to the peak time of their synthesis rate (G1: red, S: green, G2: blue, M: yellow). The numbers in brackets correspond to the numbers of genes in each group. Generally the degradation peak is shifted by approximately 20 min relative to the synthesis peak (see also Supplementary Information, Fig 37).

respective best fits, termed "variable degradation score", was used to rank genes according to their fluctuations in mRNA degradation (Supplementary Information, section 3.3). The variable degradation score was averaged over both replicate time series. Periodic transcripts had a mean variable degradation score of 0.64 ($\pm$0.47 s.d.), as opposed to non-periodic transcripts (mean 0.40, $\pm$0.44 s.d.). Conversely, genes with a variable degradation score above 0.3 comprised 74.7% of all periodic transcripts. Additionally, the variable degradation score was positively correlated with the periodicity score of periodic transcripts (Supplementary Information, Fig 38, Spearman correlation = 0.2, $P < 10^{-10}$). This indicates that periodic variation in mRNA degradation is a common feature of periodic transcripts.

We conducted a simulation study where a given time course for $\mu(t)$ was combined with periodic degradation time courses of variable amplitude to generate labeled and total mRNA profiles. Noise was added according to the MoPS error model, and the variable degradation score was calculated for all instances. The variable degradation score rose with increasing amplitude and decreasing mean of the degradation time course (Supplementary Information, section 3.4). In order to assess the power of our approach for discriminating between genes having constant respectively variable degradation rates, we calculated its sensitivity and specificity for various degradation amplitudes. The results are summarized as receiver-operating characteristic (ROC) curves (Fig 6D). We found that for transcripts with degradation score above 0.3 and with degradation amplitude above 1.5, the specificity and sensitivity of our procedure was above 87% respectively 55% (colored squares in Fig 6D). We therefore chose a conservative score cutoff of 0.3 to call genes with variable degradation. The 479 periodic genes were highly and significantly enriched for genes with variable degradation (odds ratio 3.3, *P*-value < $10^{-10}$ in a Fisher test). Changes in degradation rates might be confined to a single cell cycle phase or might be gene-specific. We grouped the 358 periodic genes with variable degradation according to the cell cycle phase in which their transcription peaks and examined the distributions of their degradation peaks (Fig 6E). It turns out that there is no specific cell cycle phase where the degra-

dation of all transcripts is maximal. Instead, there appears to be a preferential time delay between synthesis peak and degradation peak of 21 min on average (see also Supplementary Information, section 4.2 and Fig 37). To investigate the biological implication of such a delay, we performed a simulation study in which we varied the shift of the degradation peak relative to the synthesis peak while keeping all other parameters unchanged. We assessed the resulting peak height and peak time delay of the total mRNA expression time course (Supplementary Information, section 3.5). It turns out that a time delay of 20–30 min strikes an optimum balance between total RNA peak height and peak shift (Supplementary Information, Fig 33). Thus, a quantitatively high and sharp expression response can be achieved at a much lower degradation rate than for constant RNA degradation. We conclude that periodic changes in mRNA degradation rates are a common, functionally relevant property of periodically expressed genes. Periodic changes in degradation efficiently achieve a sharp peaking of mRNA expression at defined time points during the cell cycle.

## Discussion

We conducted the first systematic investigation of mRNA synthesis and degradation rates during the cell cycle, using as a eukaryotic model system the yeast *S. cerevisiae*. The cDTA mRNA labeling protocol was applied to monitor mRNA synthesis and degradation of synchronized cells along three cell cycles.

We developed MoPS, a general-purpose, model-based screening algorithm for the identification of periodic changes in time course measurements. By integrating total and labeled mRNA replicate time series, MoPS identified a reliable set of 479 genes with periodic expression during the cell cycle. Our approach is particularly robust and extracts meaningful parameters from a periodic time course. These parameters, like expression peak time, peak height, and the shape of the expression time course laid a solid basis for an in-depth analysis of the underlying biological phenomena.

   

We found that labeled and total mRNA time courses are highly similar for most genes. This indicates that transcription is the key determinant of cell-cycle phase specific mRNA expression. By clustering of the fitted gene-specific parameters of labeled expression, we identified groups of co-regulated genes. We were able to retrieve known regulatory DNA motifs and identify transcription factors that determine cell cycle phase-specific transcription, confirming and extending previous work.

The intrinsic coupling of synthesis and degradation fluctuations impedes the separation of 5′ motifs for periodic synthesis from motifs for periodic degradation. When screening for 3′ motifs, we did not find motifs that are significantly linked to variable degradation. Anyway, this seems a very difficult task, as demonstrated by a recent study (Shalem *et al*, 2013). In spite of great experimental efforts, the authors did not find any 3′ end sequence motif which explains a significant part of the variability in (steady state) mRNA degradation levels.

By comparing gene expression levels of TFs with synthesis rates of their target genes, we consistently observed that total mRNA levels of activating (repressive) cell cycle TFs peak when transcription of its targets is maximal (minimal). Further investigation of co-regulated gene clusters revealed that the timing and the magnitude of periodic expression have different causes. Genes that have common binding sites for cell cycle TFs show coherent timing of expression, but differ in their mRNA synthesis rates. Striking examples are genes exclusively regulated by MBP1, a transcription factor that has a well-studied role in regulating expression of late G1 genes. Although these genes have very similar temporal profiles, they exhibit large differences in their synthesis rates and total mRNA levels. These differences are related to the composition of the core promoter TATA sequence, and correlate with the binding of general transcription factors. Periodic genes that drive cell cycle progression or regulate fundamental processes like chromatin organization in S-phase are found to be highly induced and tend to have a consensus TATA box.

The excellent reproducibility and the high temporal resolution at which mRNA synthesis rates and total mRNA expression were determined will make our data an ideal resource for more advanced reverse engineering approaches of cell cycle related gene expression networks.

The most intriguing finding from our results is however that most periodically expressed genes show periodic changes in the degradation rates of their mRNAs. We realized that total mRNA levels peak on average only 2 min after labeled mRNA, which indicates the peak of mRNA synthesis activity. This short time delay could not be explained when constant degradation rates were assumed. Computational modeling of degradation kinetics of periodically transcribed genes indicated that the stability of mRNAs decreases shortly after transcription ceases. This highlights the importance of post-transcriptional control on the regulation of genes involved in cell cycle-associated processes. Varying mRNA degradation rates during the cell cycle were previously observed (Trcek *et al*, 2011). In this study, two mitotic periodic genes SWI5 and CLB2 show a decrease in mRNA stability after peak expression to prevent carry-over of mRNAs into the next cycle. Our results extend these findings to the majority of all periodic transcripts. It is an open question how these changes are achieved, but due to the generality of the phenomenon we suggest that increased transcript degradation following

a peak of mRNA synthesis is a passive phenomenon (Deneke *et al*, 2013). On the other hand, (Trcek *et al*, 2011) propose destabilizing, specific RNA-binding factors. Since the two hypotheses are not mutually exclusive, we expect a combination of both mechanisms. Whereas the molecular mechanisms underlying this phenomenon remain to be uncovered, our study revealed that periodic changes in mRNA synthesis and temporally delayed changes in degradation are common events that achieve concise and strong mRNA expression changes during the cell cycle.

## Materials and Methods

### cDTA of the yeast cell cycle

The *BAR1* deletion strain was generated from wt strain BY4741 (*MATa, his3Δ1, leu2Δ0, met15Δ0, ura3Δ0*) by replacing the *BAR1* open reading frame from its start- to stop- codon with a KanMX module. The Δ*bar1* strain was inoculated from a fresh overnight culture at OD600 0.1. At OD600 0.4 alpha factor (Bachem) was added at a final concentration of 600 ng/ml for 2 h. Synchronization was followed visually by counting the number of budding cells under the microscope. Cells were centrifuged for 2 min at 1,600 × *g* at 30°C and washed once with 3× the original culture volume prewarmed YPD. Cells were then resuspended in the original culture volume with prewarmed YPD. 41 consecutive samples were labeled for 5 min with 4-thiouracil every 5 min for 200 min. Labeling and sample processing was performed as described (Sun *et al*, 2012). In particular, *S. pombe* mRNA spike-ins were used as an internal standard to estimate absolute abundance of *S. cerevisae* mRNA levels in total and labeled data. FACS samples were taken for each time point, labeled with Sytox Green (Invitrogen) and processed on a FACS Calibur (Beckton Dickinson). Total RNA purification, separation of labeled RNA as well as sample hybridization and microarray scanning were carried out as previously described (Sun *et al*, 2012). The quantification of labeled and total mRNA time courses was performed in two independent biological replicates. The complete dataset is available at ArrayExpress under accession code E-MTAB-1908.

### Modeling of periodic time series by MoPS

Let $g(t_1),\dots, g(t_K)$ be a time series, e.g. of gene expression measurements, at time points $t_1,\dots,t_K$. We approximate this time series by a continuous function $\gamma(t, \Theta)$, where $\Theta$ is the set of parameters characterizing $\gamma$. We assume that the $g(t_k)$, $k = 1,\dots, K$, are measurements of the values $\gamma(t_k)$. We specify a heteroscedastic Gaussian error model that has been developed specifically to gene expression measurements (Supplementary Information, sections 1.1–1.3). According to this error model, the time series is approximated by its maximum likelihood fit $\gamma(t, \Theta)$ (Supplementary Information, section 1.5). The space F of test functions $\gamma$ used in the fitting procedure determines what we actually model — it can be periodic behavior or aperiodic behavior. In each case, the quality of fit is crucially dependent on the proper choice of F and a suitable parameterization enabling an efficient maximum likelihood search. In the MoPS algorithm, we construct periodic test functions from cosine-like functions

$$f(t; \lambda', \phi, \psi) = \cos(\psi \langle 2\pi \cdot \frac{t}{\lambda'} \rangle - \phi)$$

Here, $\lambda'$ is the cell cycle length, $\phi$ is the peak time (measured relative to the cell cycle length, which is set to $2\pi$). Additionally, the "shape" parameter $\psi$ is a bijective transformation of the interval $[0, 2\pi]$ which describes the deformation of the cosine wave (Fig 1B, Supplementary Information, section 1.3). The brackets $\langle . \rangle$ denote the remainder modulo $2\pi$.

This parameterization guarantees that the resulting functions still have exactly two periods in which they monotonically increase, respectively decrease. This parameterization is motivated by the assumption that cell cycle genes peak only once during the cell cycle. Such a parameterization automatically guards against overfitting as it forbids arbitrarily "wiggly" test functions. This is a distinctive feature of our method; other algorithms suggest the use of wavelets (Guo *et al*, 2013) or fourier base functions (Lu *et al*, 2004). The functions $f$ cannot be used directly as test functions, because they still do not account for synchrony loss, i.e. the variation of individual cell cycle lengths across the cell population. Mathematically, this means that $\lambda'$ is not a constant, but a random variable with a mean of $\lambda$ and a variance of $\sigma^2$. The observed time course of a cell population is therefore given as the integral over $\lambda'$,

$$\gamma(t; \lambda, \phi, \psi, \sigma^2) = \int f(t; \lambda', \phi, \psi) d\lambda'(\lambda, \sigma^2)$$

where the distribution $d\lambda'(\lambda, \sigma^2)$ of cell cycle lengths is modeled as a lognormal distribution with mean $\lambda$ standard deviation $\sigma$ (for a discussion on the choice of distribution, see Supplementary Information, section 1.3). Finally, the space $F$ of periodic test functions is the set of all affine transformations of these $\gamma$ functions. Conversely, we also fit an exhaustive set of non-periodic expression time courses which exhaustively represent constitutive expression, constant drift, or initial fluctuations due to the synchronization procedure. A list of all functions that are considered examples of non-periodic curves is given in (Supplementary Information, section 1.4).

**Identification of cell-cycle regulated genes**

MoPS computes a periodicity score for each gene and thus allows ranking of all genes according to their likelihood ratio to be periodically expressed resp. constantly expressed. However, there is no obvious way to assign significance to this score. We use existing knowledge derived from published studies about periodically expressed genes to define a positive set and a negative set. The positive set comprises the top 200 periodic genes from Cyclebase (Gauthier *et al*, 2008) and the negative set consists of genes that have never been classified as cell cycle regulated in any cell cycle expression study (Supplementary Information, section 2.1). The empirical distribution $f$ of all MoPS scores is fitted by a mixture of the empirical distributions $f_+$ and $f_-$ of the MoPS scores of the positive respectively the negative set,

$$f \approx \mu \times f_+ + (1 - \mu) \times f_-,$$

where the mixture coefficient $\mu \in [0, 1]$ estimates the fraction of periodic genes among all genes. Fitting of $\mu$ was done by minimization

of the Kolmogoroff-Smirnov statistic. $\mu$, $f_+$ and $f_-$ were then used to calculate the false discovery rate FDR(c) as a function of the cutoff value $c$ by

$$\text{FDR(c)} = \frac{(1 - \mu) \times \int_c^\infty f_-(t)dt}{\mu \times \int_c^\infty f_+(t)dt + (1 - \mu) \times \int_c^\infty f_-(t)dt}$$

This scheme is used in the application of MoPS to the cDTA cell cycle dataset to derive a reliable set of cell-cycle regulated genes (Supplementary Information, sections 2.1–2.3).

**Motif search, association of TFs to periodic transcripts**

Genes were grouped with *k*-means clustering ($k = 10$) according to their modeled 1 min resolution labeled expression time courses. Sequences 500 bases upstream of the respective transcription start site (SGD project, www.yeastgenome.org/download-data/sequence, Genome Release 64-1-1) were used as input for XXmotif (Hartmann *et al*, 2013) for each cluster. XXmotif was used with standard parameters, medium threshold for merging of similar motifs and set to report motifs that can occur multiple times per sequence. Motifs with an *E*-value higher than one were discarded. The positional weight matrices (PWMs) derived from ChIP-chip data (MacIsaac *et al*, 2006) and the software TOMTOM (with standard parameters and pearson correlation as comparison function) were used to assign the XXmotif found motifs to significantly similar, known TF-associated motifs (*E*-value < 1).

Subsets of the 479 periodic genes are formed by using ChIP-chip derived associations (*P*-value < 0.01) of TFs and their targets (MacIsaac *et al*, 2006): genes that are regulated by a common set of cell cycle transcription factors (of 32 TFs identified in our TF screen).

**Supplementary information** for this article is available online: http://msb.embopress.org

## Acknowledgements

We would like to thank Mai Sun, Daniel Schulz, Michael Lidschreiber, Stefanie Etzold, Ulrich Gerland, and Patrick Hillebrand for advice and discussions. PE was supported by the Deutsche Konsortium für Translationale Krebsforschung DKTK. PC was supported by the Deutsche Forschungsgemeinschaft (SFB646, TR5, SFB960, CIPSM, NIM, Leibniz-Programm), the LMUinnovativ project BIN, the Bavarian Research Center for Molecular Biosystems, an Advanced Investigator Grant of the ERC, the Jung-Stiftung, the Vallee Foundation, and DKTK. AT was supported by the Deutsche Forschungsgemeinschaft SFB680 and the BMBF e:Bio Syscore grant.

## Author contributions

KM and NP conducted cDTA time series experiments, BS performed initial data processing and normalization, PE, CD and AT developed and implemented the statistical workflow and carried out computational analyses, PC initiated the study, and DM, AT and PC supervised research, PE, PC and AT wrote the manuscript.

## Conflict of interest

The authors declare that they have no conflict of interest.

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
