## [Review Process File · Molecular Systems Biology]

Periodic mRNA synthesis and degradation co-operate during cell cycle gene expression

Philipp Eser, Carina Demel, Kerstin C. Maier, Björn Schwalb, Nicole Pirkl, Dietmar E. Martin, Patrick Cramer, Achim Tresch

Corresponding author: Achim Tresch, Max Planck Institute for Plant Breeding Research

Review timeline:

Submission date:	02 October 2013
Editorial Decision:	12 November 2013
Revision received:	09 December 2013
Acceptance letter:	16 December 2013
Accepted:	16 December 2013

Editor: Maria Polychronidou

Transaction Report:

1st Editorial Decision

12 November 2013

Thank you again for submitting your work to Molecular Systems Biology. We have now heard back from the two referees who agreed to evaluate your manuscript. As you will see from the reports below, overall the reviewers acknowledge that the study is potentially interesting. However, they raise a series of concerns, which should be carefully addressed in a revision of the manuscript.

Without repeating all the points listed below, most of the reviewers' comments refer to the need to better document and/or clarify several points throughout the manuscript. Regarding a few more fundamental issues, reviewer #2 mentions that the compatibility of the presented findings with previous works should be examined and that the advantages of the presented approach need to be clearly demonstrated. Finally, reviewer #1 suggests a number of additional analyses that would strengthen the impact of the study.

If you feel you can satisfactorily deal with these points and those listed by the referees, you may wish to submit a revised version of your manuscript. Please attach a covering letter giving details of the way in which you have handled each of the points raised by the referees.

 Reviewer #1 (Report):

The manuscript by Eser and colleagues describes the gene expression changes that occur during consecutive cell cycles of the model eukaryote *Saccharomyces Cerevisiae*. This has been done before, but what is unique about this dataset is the distinction made between synthesis and degradation of mRNAs, also coupled to an external control normalisation procedure (using spiked in

Pombe RNA) that allows proper analysis of global effects. Together, these technological steps provide a much better dataset for analysis of the mRNA changes during a eukaryotic cell cycle. The authors go on to analyze their data in various ways. Two findings stand out. The first is that upstream regulatory motifs dictate timing of synthesis whereas core promoter architecture dictate the magnitude of synthesis. The second is that cell-cycle dependent changes in mRNA degradation rates contribute to the proper cycling events. This dataset is a great improvement over previous cell-cycle gene expression datasets. The findings are also of general interest. Several improvements to the manuscript are suggested below. The first requires more analyses. The others are mainly textual changes.

1. An important avenue that the authors leave completely unexplored is what the basis is for the gene and cell cycle dependent degradation rate differences. Compared to any previous study this is really an outstanding finding. The authors should do a number of additional analyses.

a) First, they should make it clear that for different mRNAs these cell cycle dependent changes in degradation rates take place at different points in the cell cycle. This may lead to them finding out whether or not the degradation rate changes are restricted to specific mRNA in a single cell cycle phase or whether different groups of mRNAs have different cell cycle dependent changes in mRNA degradation. At the same time the authors could show that the non-cycling genes don't have changing degradation rates.

b) Second, the authors really need to do sequence analyses to see whether they can determine a sequence motif that is associated with the changes in degradation rates (for the different groups of genes analyzed above). This is really the minimal that they should do to further the mechanistic understanding. In addition, in the discussion, the authors should include a section on what is currently known about sequence dependent degradation.

In case the above analyses require more space than currently available then it would be easiest to remove the sections on transcription factor involvements that are simply repetitions of what is already known.

2. The authors discussion/explanation for the differences between the number of cycling mRNAs reported in their own study and various others is rather cursory. It would help the general understanding of cell-cycle dependent gene expression if they could be more specific about how many genes they think are really showing a cyclic pattern and why some of the previous studies (with much larger estimates) are incorrect. The way it is described in the current version leaves the impression that the number of cycling genes is between 1500 and 500.

3. It would be of general interest if the authors could specifically make a conclusion about the synthesis rates of all the other (non-cycling) genes, to make it specifically clear that these are continuously being synthesized throughout the cell-cycle. As far as I am aware, it is currently not well understood whether mRNA synthesis (for the majority of genes) simply continues throughout the cell-cycle, or shuts down for example during S-phase. It would be of interest if the authors can make a general conclusion about this.

4. The understanding of the paper would be improved if the authors were a bit more concise with regard to their use of the words gene expression, transcription and synthesis. For example, in the abstract the authors state that gene expression levels is dictated by the core promoter. Since gene expression is the final mRNA level, this is easily interpreted as meaning that the core promoter dictates the mRNA degradation rate. In the results section dealing with this the authors use the same phrase "gene expression" to mean synthesis rate even though (in my interpretation) gene expression is the combination of synthesis and degradation. My suggestion would be to use gene expression for mRNA level (as a function of synthesis and degradation), synthesis or transcription when referring to synthesis and degradation when referring to degradation. Otherwise the abstract and that part of the results/discussion section is not clear.

5. In general a doubling time of 90 minutes in yeast is about as fast as it gets (optimal media and temperature). It was therefore surprising to see a cycling time of 67 minutes. Perhaps this is related to the *bar1*-deletion? It would clarify the observations if the authors could remark on this very fast cycling time in the manuscript.

6. The role of the core promoter could be discussed in light of the simplistic model that TFIID genes

are generally thought to be house-keeping and that SAGA genes are thought to be regulated. Clearly they show that there are also subsets of TFIIID genes that are nevertheless highly regulated (in a cell-cycle dependent manner). This is an important implication of their study that is clear, but not really well concluded in their discussion.

minor points

7. The source of the transcription start site in the methods section on promoter analyses is not clearly stated.

Reviewer #2 (Report):

This paper uses the already established metabolic labeling method cDTA to derive for the first time degradation and transcription kinetics of all genes in a species during the cell cycle. Applying the method to synchronized yeast cells the authors were able to compute for each gene its change in transcription and degradation rate at each time during the cycle, to identify periodic genes at the transcription and degradation levels, and the rediscover motifs for TFs from the data. They make an important conclusion that sharp peaks are obtained by a temporal sequence of periodic mRNA synthesis and degradation maxima.

Overall I find the paper interesting, important, carefully done and very clearly written. I clearly recommend publication in MSB. My two relatively (though not absolutely) major issues relate to the novelty and the compatibility with previous works. I'm confident that after addressing these points (as well as the minor ones) that paper should be most suitable.

Major points

1. I was bothered by the lack of agreement with the de Lichtenberg periodic genes. The statement that "This showed that MoPS performed as well as state-of-the art ..." is totally unclear... if the current MoPS performs as well as others why does it give such vastly different results? Another important question is what would be the result if they run algorithms from de Lichtenberg on the new dataset! I'd ask the author to do that and report on amount of agreement between their periodic genes and the periodic genes generated by the de Lichtenberg methods, on the current dataset. The authors must provide some statement on the validity of their choice of periodic genes, since as is the reader is left confused due to lack of agreement with previous literature

2. The authors state that "We used only the labeled mRNA data in this screen, because these represent transcriptional regulation better than total mRNA profiles". This touches upon the novelty of the present study: many studies have gone through the procedure of measuring cell cycle expression, identified periodic genes and clusters and identified motifs that correspond the TFs in clusters. The premise of the present study is that it uses transcription rate rather than mere mRNA level data. But this is exactly the opportunity to examine the relative, or residual, utility of their cDTA data compared to more simple mRNA level data: do they derive better clusters and motifs from the cDTA data compared to conventional mRNA expression data. Rather than simply using the labeled and not total mRNA profile data I'd urge the authors to try demonstrate what could develop into the main premise of their work: the transcription rate data, of the type that they obtain, is indeed superior in reconstructing transcription nets; if that's not the case what's the utility of the current method??

Minor points

1. For how long synchrony is preserved? Can they estimate synchrony as a function of time?
2. I'd appreciate seeing a comparison between the two replicas of the cDTA measurement. (Fig S5 provides a measure of agreement between periodicity scores, but this is already quite processes data, while a basic synthesis rate reproducibility is missing)
3. The periodicity calculation appears to be rigorous and sound. However, I was missing a reference to the already classical approach by Tavazoie et al. (Nature Genetics 1999), who use FFT to compute the extent of cell cycle periodicity for each gene. Can the author compare the methods? A clear advantage of the current method is that it deals with synchrony loss!
4. In Fig 1C an FDR of 20% is chosen. This is rather permissive, why do the authors allow such high false discovery rate? One way to assess their false discovery rate would be to randomize the time course for each and re-do the analysis. How many genes would pass then their thresholds?
5. Why was k-means run on 10 clusters? Did they try other values of k?

Reviewer #3 (Report):

The manuscript presents the genome-wide determination of mRNA levels and synthesis rates in cell cycle synchronized cells of the budding yeast *S. cerevisiae* (and, indirectly, mRNA decay rates). The data are then extensively mined to identify periodically expressed genes, potential transcriptional regulators, and effects of mRNA decay levels on mRNA temporal profiles.

This is an excellent piece of work. The experimental work is novel - this is the first time that synthesis rates have been determined during the cell cycle of a eukaryotic organism. The data are analyzed comprehensively and carefully, and the paper is generally well presented. I only have a few minor comments about issues that are unclear in the paper. There are no page numbers (should be included in any revised version), so I am numbering the pages starting with the title page as page 1.

1. Page 4 - 'principled' should be 'principal'
2. Page 7 /figure 1B. Figure 1B gives the impression that the parameter sigma is measuring a difference in amplitude, when in fact its units are time. This is confusing.
3. I don't understand what is plotted on figure 1A. Is this a particular gene or group of genes?
4. Page 9. The authors note that the mean expression level for perfect TATA-containing genes of periodic genes is higher compared to genes with mismatches to the TATA consensus. However, this also seems to be the case for non-periodic genes. A significance test is only presented for periodic genes. If the difference is not significant for non-periodic genes, this should be presented. Otherwise, the discussion of the figure should be modified to accommodate this fact.
5. Legend to figure 1 - should it be 'sigma' and not 'sigma squared'?

1st Revision - authors' response

09 December 2013

We thank you and the reviewers for the timely review of our manuscript and the overall very positive feedback. In our revised manuscript, we carefully took into account the reviewers' comments, which we found instructive and helpful. All relevant changes in the main text and in the Supplements are marked in red. Please find below our point-to-point response to the reviewers. We are looking forward to your decision.

Reviewer #1:

- *The manuscript by Eser and colleagues describes the gene expression changes that occur during consecutive cell cycles of the model eukaryote *Saccharomyces Cerevisiae*. This has been done before, but what is unique about this dataset is the distinction made between synthesis and degradation of mRNAs, also coupled to an external control normalisation procedure (using spiked in *Pombe RNA*) that allows proper analysis of global effects. Together, these technological steps provide a much better dataset for analysis of the mRNA changes during a eukaryotic cell cycle. The authors go on to analyze their data in various ways. Two findings stand out. The first is that upstream regulatory motifs dictate timing of synthesis whereas core promoter architecture dictates the magnitude of synthesis. The second is that cell-cycle dependent changes in mRNA degradation rates contribute to the proper cycling events. This dataset is a great improvement over previous cell-cycle gene expression datasets.*

The findings are also of general interest. Several improvements to the manuscript are suggested below. The first requires more analyses. The others are mainly textual changes.

We thank the reviewer for the appreciation of our findings.

1. *An important avenue that the authors leave completely unexplored is what the basis is for the gene and cell cycle dependent degradation rate differences. Compared to any previous study this is really an outstanding finding. The authors should do a number of additional analyses.*

a) *First, they should make it clear that for different mRNAs these cell cycle dependent changes in degradation rates take place at different points in the cell cycle. This may lead to them finding out whether or not the degradation rate changes are restricted to specific mRNA in a single cell cycle phase or whether different groups of mRNAs have different cell cycle dependent changes in mRNA degradation. At the same time the authors could show that the non-cycling genes don't have changing degradation rates.*

This is a very useful suggestion, which led us to replace the former Figure 6D with the new Figure 6E, in which the distribution of degradation peak times is shown. The degradation peaks are not confined to a certain phase of the cell cycle. They appear to have a preferential time delay of 21min relative to the synthesis peak (see also Supplements 4.2). We have also added an even more detailed scatter plot comparing the synthesis and degradation peak times of periodic genes with variable degradation (Supplementary Figure S37). The Results paragraph on periodic changes in mRNA degradation has been extended by a discussion of these new results.

As for the non-cycling genes, we refer to our answer to the third point.

b) *Second, the authors really need to do sequence analyses to see whether they can determine a sequence motif that is associated with the changes in degradation rates (for the different groups of genes analyzed above). This is really the minimal that they should do to further the mechanistic understanding. In addition, in the discussion, the authors should include a section on what is currently known about sequence dependent degradation.*

In case the above analyses require more space than currently available then it would be easiest to remove the sections on transcription factor involvements that are simply repetitions of what is already known.

We share the desire to link periodic degradation to sequence motifs. Of course we did search for “periodic degradation motifs”. As a result, we recovered 5'-motifs already known to cause periodic mRNA expression. All motifs found could be related to cell-cycle transcription factors; these are presumably involved in mRNA synthesis regulation rather than in mRNA degradation. While at first sight this seems disappointing, it is not unexpected. Since we show that essentially all periodic transcripts have a fluctuating degradation rate, it is logically impossible to separate motifs for periodic synthesis from motifs for periodic degradation. Unfortunately, we did not find sequence motifs at the 3' end of the transcripts with periodic degradation. This is in accordance with a very recent publication where the authors do not find any 3' end sequence motif which explains a significant part of the variability in (steady state) mRNA degradation levels (Shalem et al., PLoS Comp Biol 2013;9(3)). We have included the above reference in the text. In the discussion, we highlight the intrinsic coupling of periodic degradation and that periodic synthesis precludes a distinction between motifs relevant for mRNA synthesis and motifs relevant for mRNA degradation.

2. *The authors discussion/explanation for the differences between the number of cycling mRNAs reported in their own study and various others is rather cursory. It would help the general understanding of cell-cycle dependent gene expression if they could be more specific about how many genes they think are really showing a cyclic pattern and why some of the previous studies (with much larger estimates) are incorrect. The way it is described in the current version leaves the impression that the number of cycling genes is between 1500 and 500.*

There is no clear dichotomy between periodically expressed genes and constant genes. Hence, the number of genes called ‘periodic’ is mostly a consequence of the stringency cutoff and the screening method used (we say this in paragraph 2 of section “Characterization of periodically expressed genes”). This was reason enough for us to develop our own, very reliable periodicity screen. To illustrate the gradual transition from periodically to constantly expressed genes, we have added a

supplementary figure (S9) which shows representative samples of genes with various periodicity scores.

All of the previous studies cited in the manuscript produced high-quality data, and they used sound methods to call periodic transcripts. Therefore, it is not justified to claim superiority of one method (respectively of one estimate of the number of cycling genes). The more interesting question, namely which of these fluctuations is biologically relevant, remains open, and its systematic investigation is not the scope of our work.

3. It would be of general interest if the authors could specifically make a conclusion about the synthesis rates of all the other (non-cycling) genes, to make it specifically clear that these are continuously being synthesized throughout the cell-cycle. As far as I am aware, it is currently not well understood whether mRNA synthesis (for the majority of genes) simply continues throughout the cell-cycle, or shuts down for example during S-phase. It would be of interest if the authors can make a general conclusion about this.

This is an excellent suggestion! In the end we were using only 10% of the data when focusing exclusively on the 479 genes that are called periodic. Your recommendation led to two substantial improvements in the manuscript:

First, to answer your question about global fluctuations of gene expression, we performed another periodicity screen on labeled data, now using only constant functions as non-periodic test-functions. Among the reliable genes (genes above a certain minimum expression level), we keep 500 genes with the *lowest* periodicity scores, i.e., the genes that are most constantly expressed. After normalizing their gene profiles to mean expression 1 and visualizing the normalized labeled expression profiles, the average time course of the 500 most constant genes provides a lower bound for the global fluctuations in mRNA synthesis (Supplementary Figure S23). We do not find any evidence for a global mRNA synthesis stop in S-phase. This might be due to the fact that replication occurs at multiple origins, making it an asynchronous (local) event for different genes. Duplication of a gene takes only a few minutes, thus synchrony decay will blur this event in our cell population average.

Second, by ignoring the time at which measurements were taken, we use all labeled and total measurements of one mRNA as replicates to calculate a high precision estimate of its (cell-cycle averaged) synthesis and degradation rate. We compared these estimates with the up to date most recent estimates in (Sun et al., Genome Res. 2012) obtained by the same cDTA technique. Encouragingly, they are in excellent agreement not only on a relative, but also on absolute scale (Supplementary Figure S7). The high number of replicates allowed us to additionally derive an empirical variance estimate. We have added supplementary tables with our new, probably most accurate genome-wide estimates of mRNA synthesis rates, half lives, and their variances. A mention of it is included in the main text.

4. The understanding of the paper would be improved if the authors were a bit more concise with regard to their use of the words gene expression, transcription and synthesis. For example, in the abstract the authors state that gene expression levels is dictated by the core promoter. Since gene expression is the final mRNA level, this is easily interpreted as meaning that the core promoter dictates the mRNA degradation rate. In the results section dealing with this the authors use the same phrase "gene expression" to mean synthesis rate even though (in my interpretation) gene expression is the combination of synthesis and degradation. My suggestion would be to use gene expression for mRNA level (as a function of synthesis and degradation), synthesis or transcription when referring to synthesis and degradation when referring to degradation. Otherwise the abstract and that part of the results/discussion section is not clear.

We apologize for being sloppy in a few cases, the intended use of “synthesis” and “mRNA expression” is exactly as suggested by the reviewer. We have corrected this wherever necessary.

5. *In general a doubling time of 90 minutes in yeast is about as fast as it gets (optimal media and temperature). It was therefore surprising to see a cycling time of 67 minutes. Perhaps this is related to the bar1-deletion? It would clarify the observations if the authors could remark on this very fast cycling time in the manuscript.*

To be precise, we determined the average cell cycle time as 62.5 min (Figure 2A). The bar1-deletion does not influence the doubling time, since this mutation does not change growth and doubling time in unsynchronized cells. Our estimated mean cell cycle time is comparable to that in a recent study (Granovskaia et al., Genome Biology 2010) and the study by Spellman et al. (Mol Biol Cell, 1998) where they also used alpha factor for the synchronization of *S.cerevisiae*. In both studies a doubling time of approx. 65 minutes is observed. We have added this remark in the main text.

6. *The role of the core promoter could be discussed in light of the simplistic model that TFIID genes are generally thought to be house-keeping and that SAGA genes are thought to be regulated. Clearly they show that there are also subsets of TFIID genes that are nevertheless highly regulated (in a cell-cycle dependent manner). This is an important implication of their study that is clear, but not really well concluded in their discussion.*

In Figure 5C we show the weak correlation of TFIID occupancy with mean expression levels of non-periodic and particularly with periodic genes. We examined TFIID occupancy of periodic genes and found their occupancy levels below average. We therefore think that TFIID plays, if at all, only a minor role in regulating cell-cycle genes.

minor points

7. *The source of the transcription start site in the methods section on promoter analyses is not clearly stated.*

We used the transcription start site annotation from the SGD database. This is now mentioned in the Methods.

Reviewer #2:

- This paper uses the already established metabolic labeling method cDTA to derive for the first time degradation and transcription kinetics of all genes in a species during the cell cycle. Applying the method to synchronized yeast cells the authors were able to compute for each gene its change in transcription and degradation rate at each time during the cycle, to identify periodic genes at the transcription and degradation levels, and the rediscover motifs for TFs from the data. The make an important conclusion that sharp peaks are obtained by a temporal sequence of periodic mRNA synthesis and degradation maxima.

Overall I find the paper interesting, important, carefully done and very clearly written. I clearly recommend publication in MSB. My two relatively (though not absolutely) major issues relate to the novelty and the compatibility with previous works. I'm confident that after addressing these points (as well as the minor ones) that paper should be most suitable.

Thanks, we feel encouraged by the appreciation of our work.

Major points

1. I was bothered by the lack of agreement with the de Lichtenberg periodic genes. The statement that "This showed that MoPS performed as well as state-of-the art ..." is totally unclear...

The full sentence in our paper says "This showed that MoPS performed as well as state-of-the art methods for the identification of periodically expressed genes (Supplements S2.9 and Figure S25)." In Supplements S2.9, we performed an extensive validation study in which we compare MoPS with the results of the de Lichtenberg "gold standard" method and 6 other published methods and datasets. The convincing performance of our method is shown in Supplemental Figure S25. It is true that our statement becomes clear only when considering the Supplements. We initially thought of including Figure S25 into the main text, but decided against it for the sake of presenting biologically

relevant results. We would like to keep this unchanged, but we added information on how performance was assessed (by a receiver operating characteristic analysis) to the main text.

... if the current MoPS performs as well as others why does it give such vastly different results? A similar question has been asked by reviewer 1 (second point). We therefore refer to our answer there. Additionally, let us say that the agreement between the different sets is nevertheless highly significant ($p < 10^{-5}$ in all pairwise Fisher tests). Maybe our statement "The agreement between the datasets/methods is poor (de Lichtenberg et al, 2005a) (Supplemental Figure S17)" was misleading. The agreement was admittedly lower than we hoped, but the agreement of our periodic gene set with other such sets was not lower than the agreement between any two other of these sets. We have reformulated the above sentence and mention the significant overlap of the gene sets, to avoid confusion.

Another important question is what would be the result if they run algorithms from de Lichtenberg on the new dataset! I'd ask the author to do that and report on amount of agreement between their periodic genes and the periodic genes generated by the de Lichtenberg methods, on the current dataset. The authors must provide some statement on the validity of their choice of periodic genes, since as is the reader is left confused due to lack of agreement with previous literature.

Unfortunately, the de Lichtenberg method does not exist as a publicly available implementation. We think that our validation study in Supplements S2.9 is extensive enough to validate MoPS. Please bear in mind that our primary goal was not the development of a screening procedure which outperforms all others in terms of sensitivity and specificity in a certain benchmark test. We rather wanted to develop a model-based screening whose parameters have an intuitive biological meaning. The results of MoPS are immediately accessible to biological interpretation, which is of immense practical value. To emphasize this further, we have added a corresponding statement at the end of the "Characterization of periodically expressed genes" paragraph.

2. The authors state that "We used only the labeled mRNA data in this screen, because these represent transcriptional regulation better than total mRNA profiles". This touches upon the novelty of the present study: many studies have gone through the procedure of measuring cell cycle expression, identified periodic genes and clusters and identified motifs that correspond to TFs in clusters. The premise of the present study is that it uses transcription rate rather than mere mRNA level data. But this is exactly the opportunity to examine the relative, or residual, utility of their cDTA data compared to more simple mRNA level data: do they derive better clusters and motifs from the cDTA data compared to conventional mRNA expression data. Rather than simply using the labeled and not total mRNA profile data I'd urge the authors to try demonstrate what could develop into the main premise of their work: the transcription rate data, of the type that they obtain, is indeed superior in reconstructing transcription nets; if that's not the case the what's the utility of the current method??"

In our present manuscript, we exploit the differences in labeled and total mRNA measurements in two ways: 1) We use the labeled expression of identified TFs that act as periodic activators or periodic inhibitors together with the total expression of their targets (Figure 4), 2) We observe surprisingly small differences in total and labeled peak times (Figure 6B). The latter lead us to the major discovery of global fluctuations in mRNA degradation of periodic transcripts. We have added a statement emphasizing the utility of the cDTA technique to the Discussion.

The reviewer's suggestion to use a combination of labeled and total data for the reconstruction of cell-cycle related transcription networks is great. Indeed, we are currently preparing another manuscript which is exclusively dedicated to this topic. We can safely say that it is a separate story that exceeds the scope of this manuscript. We now mention the opportunity for an improved transcription network reconstruction in the Discussion.

Minor points

1. For how long synchrony is preserved? Can they estimate synchrony as a function of time?

The loss of synchronization is modeled by the “synchrony loss” parameter σ (see Figure 1B). The Supplemental sections S1.4 and 1.5 give a formal definition of this parameter, which does not change with time. The synchrony loss describes the standard deviation of the distribution of cell cycle lengths in our cell population. We included a comparison of different distributions for modeling the loss of cell synchronization in the Supplements (Supplements S1.3, Figure S2). It demonstrates that the variance σ^2 (respectively the parameter σ which we call synchrony loss) of these distributions is the only relevant parameter. The result of the synchrony loss estimation for our data is shown in Figure 2A. Furthermore, we extended Figure 2B (bottom panel) and the corresponding legend to illustrate the cell cycle time distribution at each time point in the first two cycles.

The synchrony of the cell population (i.e., the distribution of the cell cycle times in the cell population) at each time point is stored in the kernel function k , which is shown in Supplementary Figure S3. We added a reference to Supplementary Figure S3 in the main text.

2. I'd appreciate seeing a comparison between the two replicas of the cDTA measurement. (Fig S5 provides a measure of agreement between periodicity scores, but this is already quite processed data, while a basic synthesis rate reproducibility is missing)

This is a valid point. The measurements in each replicate are highly correlated between replicates in total and labeled datasets. The mean Pearson correlation of measurements between replicates is 0.97 (min=0.93, max=0.99). We followed the reviewer's advice and added supplementary figures (Figure S4 and S5). Figure S4 contains a histogram of the gene-wise relative errors between two replicates for all samples, and it shows a scatter plot for one replicate time point, confirming excellent reproducibility. Figure S5 shows (separately for the labeled and the total RNA measurements) the pairwise correlations between all time points in both replicates. It nicely demonstrates that the majority of the variation between samples is due to biological variation, namely periodic gene expression, and not due to noise. One clearly recognizes three diagonal bands of high correlation showing that samples taken at 60-65min (=one average cell cycle length) intervals correlate best. This holds for labeled and for total mRNA measurements. Also, by correlating labeled and total measurements, we demonstrate once more that the time delay between labeled and total expression is surprisingly short, corroborating our findings in Figure 6B.

3. The periodicity calculation appears to be rigorous and sound. However, I was missing a reference to the already classical approach by Tavazoie et al. (Nature Genetics 1999), who use FFT to compute the extent of cell cycle periodicity for each gene. Can the author compare the methods?

A clear advantage of the current method is that it deals with synchrony loss!

We mention the approach by Lu et al. (2004), which use a Fourier decomposition of the time series in a statistically elegant and rigorous way. We have additionally included the paper by Tavazoie et al. to our references.

4. In Fig 1C an FDR of 20% is chosen. This is rather permissive, why do the authors allow such high false discovery rate? One way to assess their false discovery rate would be to randomize the time course for each and re-do the analysis. How many genes would pass then their thresholds?

Randomization of the time points is a good idea for controlling the type-I error rate of the periodicity test. The FDR is more informative than the type-I error rate for controlling the expected number of truly periodic genes among the predicted ones, though. There are methods for controlling the FDR in ignorance of the periodicity score distribution under the hypothesis of periodic expression (Rustici et al. Nature Genetics 2004, Lu et al. NAR 2004). However, our procedure exploits the existence of a *bona fide* set of periodically expressed genes [Lichtenberg et al.], thus producing a more reliable estimate of the FDR. Believing that we have obtained a realistic FDR estimate, 20% false positives seems a reasonable threshold. The validation by the de Lichtenberg

benchmark (Supplements S2.9) provides an additional post hoc justification of this choice. Therefore, we would like to keep the screening procedure as it is.

5. Why was k-means run on 10 clusters? Did they try other values of k?

Initially, we performed the clustering with $k = 1, 2, \dots, 100$ clusters and calculated for each clustering the within-cluster sum of squares (see the plot below). This measure drops with increasing k and it is a common good practice to use a k in the range where the curve has a 'knee'. We thus used and exploited k-means clustering with k in the range of 8 to 15. Upon visual inspection of the clustered expression profiles and comparison of the resulting motifs we decided to use $k = 10$.

Reviewer #3:

- The manuscript presents the genome-wide determination of mRNA levels and synthesis rates in cell cycle synchronized cells of the budding yeast *S. cerevisiae* (and, indirectly, mRNA decay rates). The data are then extensively mined to identify periodically expressed genes, potential transcriptional regulators, and effects of mRNA decay levels on mRNA temporal profiles.

This is an excellent piece of work. The experimental work is novel - this is the first time that synthesis rates have been determined during the cell cycle of a eukaryotic organism. The data are analyzed comprehensively and carefully, and the paper is generally well presented.

We thank the referee for his positive assessment.

- I only have a few minor comments about issues that are unclear in the paper. There are no page numbers (should be included in any revised version), so I am numbering the pages starting with the title page as page 1.

1. Page 4 - 'principled' should be 'principal'

This has been corrected.

2. Page 7 /figure 1B. Figure 1B gives the impression that the parameter σ is measuring a difference in amplitude, when in fact its units are time. This is confusing.

We thank the referee for this useful remark. Indeed, the synchrony loss σ is the only parameter which cannot be read off directly from Figure 1B. We modified Figure 1B; the bracket at the right

hand side now reads “attenuation due to synchrony loss σ “. We now describe the parameter σ in more detail in the paragraph “model-based periodicity screening”.

3. I don't understand what is plotted on figure 1A. Is this a particular gene or group of genes?
The four time courses shown in Figure 1A correspond to the two replicates of the labeled respectively total mRNA measurements of the URH1 (YDR400W) gene. We have added an explanation in the figure legend.

4. Page 9. The authors note that the mean expression level for perfect TATA-containing genes of periodic genes is higher compared to genes with mismatches to the TATA consensus. However, this also seems to be the case for non-periodic genes. A significance test is only presented for periodic genes. If the difference is not significant for non-periodic genes, this should be presented. Otherwise, the discussion of the figure should be modified to accommodate this fact. Indeed, for non-periodic genes the mean expression level for perfect TATA genes is significantly higher than for those with TATA-mismatches (Wilcoxon test $p < 1.7 \times 10^{-5}$). However, the difference between the mean values is much higher for periodic genes (320 transcripts per cell for the group with 1 mismatch in TATA / 330 transcripts per cell for the group with 2 mismatches) than for non-periodic genes (100 respectively 110 transcripts per cell). We clarified this in the main text.

5. Legend to figure 1 - should it be 'sigma' and not 'sigma squared'?

The synchrony loss parameter we are interested in is σ , we have corrected this. The mistake resulted from the habit to report the variance of a distribution – the synchrony loss is in fact chosen such that σ^2 is the variance of a suitable distribution (see Supplements S1.3, Equation (9), and (10)-(13)).

Acceptance letter

16 December 2013

Thank you again for sending us your revised manuscript. We have now heard back from the referee who accepted to evaluate your revised manuscript. As you will see, the referee is now satisfied with the modifications made. I am therefore pleased to inform you that your paper has been accepted for publication.

Thank you very much for submitting your work to Molecular Systems Biology.

Reviewer #1:

The authors have responded adequately to the referee reports.